# Improving Customisation in Clinical Pathways by Using a Modular Perspective

**DOI:** 10.3390/ijerph182111129

**Published:** 2021-10-22

**Authors:** Anne van Dam, Margot Metz, Bert Meijboom

**Affiliations:** 1McCoy & Partners, Torenallee 45, 5617 BA Eindhoven, The Netherlands; anne.van.dam@mccoy-partners.com; 2GGz Breburg, Specialist Mental Health Care Organisation, P.O. Box 770, 5000 AT Tilburg, The Netherlands; 3Scientific Centre Tranzo, Tilburg School of Social and Behavioural Sciences, Tilburg University, P.O. Box 90153, 5000 LE Tilburg, The Netherlands; b.r.meijboom@tilburguniversity.edu; 4Department of Management, Tilburg University, P.O. Box 90153, 5000 LE Tilburg, The Netherlands; 5Department of Marketing, Innovation and Organization, Ghent University, Tweekerkenstraat 2, 9000 Ghent, Belgium

**Keywords:** shared decision making, patient-centred care, evidence-based practice, clinical guidelines, modularity, specialist mental health care

## Abstract

A standardised system of clinical pathways often conflicts with providing patient-centred heterogeneous care. Mental health care organisations are searching for new methods to become responsive towards unique treatment needs. Modularity is a method increasingly suggested to reconcile standardisation and customisation. The aim is to investigate the extent to which modularity can be applied to make clinical pathways in specialist mental health care more flexible in order to stimulate shared decision making (SDM) and thereby customise care processes to patient contexts while maintaining evidence-based standards. Methods consist of literature research and a theory-based case study including document analysis and semi-structured interviews, which were performed at a Dutch specialist mental health care organisation. The results show that in current literature two modularity-based structures are proposed that support flexibility and customisation, i.e., ‘Prototype’ and ‘Menu-based’. This study reveals that departments tend to use the prototype method if they have predictable patient needs, evidence-based methods are available and there is sequency in treatment components. The menu-based method is preferred if there are unpredictable needs, or the evidence needed to create interconnectedness in treatment is lacking. In conclusion, prototype or menu-based methods are both suitable for applying SDM and reaching customisation in practice. The choice is determined by three characteristics: predictability of needs, availability of evidence and the interconnectedness of treatment components.

## 1. Introduction

The clinical pathways approach is an increasingly used method of organising care. Around the early 1990s, it was implemented for the first time in the UK [1]. According to the European Pathways Association, it can be defined as a method “for the mutual decision making and organisation of care processes for a well-defined group of patients during a well-defined period” [2]. Its main goals are to make more efficient use of resources and improve quality by increasing the use of evidence-based guidelines in practice [1,3]. Clinical pathways are used to organise care for a homogeneous patient group with relatively clear demands [3]. Moreover, clinical pathways are only suitable processes involving treatments that can be represented as a linear sequence [4]. Therefore, the use of clinical pathways as a standardised system may conflict with the requirement for patient-centred heterogeneous care [5]. Clinical pathways do not target patients with unique requirements and contexts [6] and do not facilitate shared decision making (SDM).

In this paper, we study the extent to which flexibility in the use of clinical pathways in specialist mental health care can stimulate SDM and thereby enhance patient-centeredness. SDM is defined as the collaborative approach in which patients, together with their relatives, and practitioners share available information about choices in treatment from both perspectives and patients are supported in participating actively in decision making about treatment [7].

In order to achieve a higher level of patient-centredness in combination with evidence-based treatment, practitioners, patients and their relatives are encouraged to adopt SDM. However, in clinical practice the application of SDM is lagging behind because it requires another way of working and additional efforts at the individual and organisational level. At the individual level, it requires changes in roles at all sides of the dyad (practitioners, patients and relatives), and at the organisational level adjustments in the workplace culture and organisation of treatment are required [8,9,10,11,12,13]. An important aspect of the organisation of treatment is that it calls for a flexible approach that challenges the routine of working with clinical pathways. Today, it is important to give treatment according to up-to-date evidence and practice-based guidelines, in an efficient way, based on information that is clear and comprehensible for patients and also tailored to their preferences and needs. These aspects require a lot of flexibility and transparency in the organisation of treatment and the way clinical pathways are used in daily practice [8,12,14]. Moreover, patients should be provided with better information (regarding their treatment options) to facilitate SDM [13].

Modularity could possibly provide an answer to the challenges mentioned above. It is a concept originating from operations management [15] that is increasingly recommended to reconcile standardisation and customisation in order to meet the heterogeneous demands of patients [16]. Modularity refers to a set of principles for managing complexity by dividing a system into standardized parts in order to create a variety of configurations to meet heterogeneous needs [17,18]. Though modularity in services is getting more attention in current publications [19,20], research on the application of modularity in health services remains scarce [21,22,23,24], especially in the multidisciplinary context of specialist mental health care. In this field, there has been a recent increase in attention to a holistic and flexible approach in treatment planning in line with patient needs, evidence and practice-based knowledge and based on decision making with patients and relatives [19,20,21,22].

The goal of this research is to investigate the extent to which modularity can be applied to make clinical pathways in specialist mental health care more flexible in order to stimulate SDM and thereby customise the care process while maintaining quality standards and evidence-based guidelines.

## 2. Materials and Methods

This cross-sectional research comprised a literature study followed by a case study performed at a Dutch specialist mental health care organisation. Thus, this Dutch specialist mental health care organisation, with diverse patient groups and treatment departments, was the study case. For the literature study, a search was conducted in PubMed and Google Scholar based on the following keywords: “modularity”, “modularity in (mental) health care provision”, “clinical pathways”, “personalisation” and “customisation (in health care”, “(mental) health care provision”, “patient-centeredness”, “shared decision making”, and combinations of those keywords. The researchers focused on articles published between 2000 and 2020. The references of these articles were checked for additional potentially relevant studies. Initially, 75 articles were deemed relevant based on their titles and abstracts. This search yielded the background and basis of this article, which were initially assessed on the basis of titles and abstracts. Ultimately, 30 articles containing the relevant (combinations of) subjects were selected for further analysis.

Currently, research and theory in this area are in their formative stages. For this reason, a descriptive case study was particularly appropriate. Variables are still unknown and the phenomenon to be investigated is not well understood. With this cross-sectional, descriptive case study, the intention was to gain a better understanding of the study population and their insights on the topic as a first step. The case study consisted of document analysis and semi-structured interviews with heterogeneous stakeholders, i.e., representatives of the patient council and practitioners treating different patient groups, based on themes from the literature study, the research question and document analyses. Document analysis consisted of analysing existing reports and minutes of relevant meetings. After the interviews, the audio recordings were transcribed and de-identified to ensure the confidentiality of data by pseudonymisation. Interviews were conducted for one month to represent the current situation in a particular month in time. The sample size was based on a balanced distribution of practitioners working in different treatment departments. Moreover, interviewees were included until a point of data saturation was reached. By using different data sources, i.e., interviews, literature and document analysis, and by engaging multiple researchers to analyse the data, data triangulation was facilitated and hereby the reliability of results improved.

A total of 23 representatives consisting of 2 representatives of the patient council, 14 practitioners treating different patient groups of all ages (i.e., depression, bipolar, anxiety, eating, personality psychotic disorders), 3 managers and 4 representatives of supportive services participated in the interviews. The interviews were supported by a semi-structured interview protocol using a topic list. This topic list was compiled based on the results of the literature study and document analysis, and it addressed the research question. It was also translated towards a coding scheme. In this way, literature was used to study the empirical reality. The topic list addressed the following topics: current use of clinical pathways, the role and involvement of patients, standardisation, and modular organisation. The following are examples of questions that were asked during the interview: ‘What is the role of the patient in choosing a clinical pathway?’, ‘To what extent are you flexible in the care you offer (i.e., content, input, how)?’, ‘Would it be possible to divide the current ‘basic’ care packages/clinical pathways into modules (so that it would result in the menu-based method)?’ In addition, the interviews focused on the two customization methods previously described (prototype and menu-based).

Data coding of the transcripts and documents was conducted by a thematic analysis. The researchers started with deductive coding, based on the literature study and research question, and then performed inductive coding. Two researchers (*MM*, *AvD*) independently coded the transcripts of the first three interviews. They then discussed the coding to reach consensus in order to increase the robustness and reliability of the results. The agreed thematic codebook was then used by one researcher (*AvD*) to code subsequent transcripts.

## 3. Results

### 3.1. Literature Study

The literature study reveals two appropriate modularity-based approaches that support flexibility and customisation: ‘Prototype’ and ‘Menu-based’ [25]. In the first approach, a customer (in health care: patient) gets a (standardized) prototype of treatment components which can be further tailored to suit their requirements. To meet specific needs, components of the prototype are modified, or modules are created, deleted or added. Therefore, here customisation is realised by changing dimensions of the prototype. This method, Method I, is visualized in Figure 1. One can see that the standardized prototype could be adjusted with different modules.

In the second approach, modularity is applied via a menu of options from which a service employee combines several components in order to meet the specifications of the customer [25,26]. Thus, customisation is created by (re-)combining menu components from an arranged set of standardized components. This approach is visualised in Figure 1, Method II. In health care, a patient should be able to select suitable modules together with the practitioner. Within both methods, modules from other organisations can be used.

In mental health care, patients often prefer to be involved in decision making about their treatment [11,27,28], and this is called shared decision making (SDM).Applying SDM improves patient-centred care and contributes to the customisation of treatments. To foster SDM, clinical pathways should be flexible and simplified. In the original standardised system of clinical pathways, they were not flexible or simplified [25] but the prototype and menu-based methods may improve this. It is important to help patients and practitioners apply SDM in daily practice, so that patients actually get the chance to participate actively and feel competent doing so. Influencing factors that contribute to applying SDM in clinical practice are the following: comprehensible information and explanation to patients about treatment options, clarifying their own values which are important in choosing the best suitable option, and experiencing support and flexibility during decision making. The prototype and menu-based methods contribute to these factors, and facilitate patients and practitioners in the decision making about treatment [11].

### 3.2. Case Study

The document analysis and interviews reveal that positive features of using clinical pathways are integrated quality, guidance, and a clear framework of processes. As one practitioner (Pr.) mentioned during one of the interviews:

*“So, it is treatment, proper treatment, according to clinical pathways, with proper methods, (…) with high-quality personnel, who make sure that it can be done in the best possible way. It is not more or less than good quality for patients.”* (Pr. 1)

Consequently, patients can expect high-quality care and trust the experience of the practitioner. It is emphasized that, especially at the beginning of a treatment process, a kind of guidance derived from evidence is helpful. The main disadvantage is the lack of flexibility because of the level of detail in which pathways in this case were described. For example, in clinical pathways treatment duration, the content and number of sessions of the whole treatment process were already determined in advance. In this organisational case, each patient always had to be assigned to one pathway, which hinders SDM and customisation. One practitioner described the sometimes-unsuitable application of clinical pathways:

*“What we do now: we just look at what fits best, what does this resemble the most? Then we put the patient on that clinical pathway. It is often not correct at all.”* (Pr. 4)

Moreover, another identified drawback that results from the rigid descriptions is that there are no treatment options to discuss. This makes SDM harder to apply.

However, the needs of some patient groups could be too unpredictable to fit in any prescribed pathway at all or for that patient group evidence about the sequence in treatment is lacking.

Both methods, prototype and menu-based, were thoroughly evaluated during the interviews (Table 1). The results indicate that both methods are appropriate to provide transparency and flexibility to patients. Representatives highlighted that standardization should be integrated in the way of working. In general, representatives admitted that transparency to patients about the content of clinical pathways can be improved. Interviewees stated that it is important to be clear about the treatment content. More information about the content and options of modules would provide better involvement of patients by opening up the conversation:

*“It would be attractive if you had this (Method II, Menu-based), so that patients would know which modules exist. Then you would talk more about that.”* (Pr. 6)

This is underlined by a patient representative (Pa.):

*“What does it all involve? That is what I would have liked to know afterwards”* (Pa. 1)

With the provided information, patients, practitioners and relatives could think together about what best fits the situation. This could enhance SDM.

The interviews also reveal that the prototype method entails the same benefits as the current clinical pathways but offers options to customise them and creates flexibility. Method II seems to offer more customisation possibilities compared to Method I, but results in less overview. A parallel is made by one practitioner:

*“I see this more as, for example, a fast-food chain. They have also put together menus for you there, because often that is also based on choices that people have made before.”* (Pr. 4)

For Method I, almost all practitioners argued that it should be possible to add modules of another care centre or remove modules from the prototype. For Method II, interviewees argued that it should be possible to add treatments of other care centres or institutions.

Figure 2 represents a conductive scale on which the preference for a certain method is outlined. Three points are detected that determine the preference for a certain method: the availability of knowledge, the predictability of treatment needs, and the interconnectedness between treatments. If evidence and practice-based knowledge about how to fill in a clinical pathway exists, practitioners prefer Method I ‘Prototype’. This is in contrast to the situation when there is less evidence and practice-based knowledge about how to set up a clinical pathway. Moreover, if patient needs are predictable, then practitioners prefer Method I. When patient needs are less predictable and thus practitioners cannot predict what the next treatment (characterized as module) will be, they prefer Method II. The last characteristic that determines the preference for a certain method is the interconnectedness (sequence) in treatments. If a logical treatment sequence exists, i.e., when treatment B usually follows treatment A, practitioners prefer Method I. When no interconnectedness or sequence in treatments exists, practitioners choose Method II. Both methods have sufficient flexibility to facilitate SDM when making treatment decisions.

## 4. Discussion

### 4.1. Main Findings

The literature study revealed two modularity-based methods, prototype and menu-based. Compared to the original clinical pathways system, both methods support flexibility and customisation of treatment. With the prototype method, a patient gets a (standardized) prototype of treatment components which can be further adjusted to suit to the patient’s treatment needs. To meet specific needs, components of the prototype can be created, deleted or added. In the second approach, the menu-based method, customisation is created by (re-)combining components from an arranged set of standardized components.

The case study showed that the suitability of these two methods is a conductive scale based on predictability of patients’ needs, sequence or interconnectedness of activities within treatment, and availability of evidence to create a prototype. To increase flexibility in the prototype method, it should be possible to add or remove treatment modules. Both methods are sufficiently flexible to be suitable for facilitating SDM between patients, relatives and practitioners regarding treatment planning, especially if you can combine them with treatment modules from other departments.

### 4.2. Interpretation and Clinical Implications

The availability of evidence-based knowledge, the predictability of needs, and interconnectedness between treatments of the target group determine the preference for a certain modularisation method. Both methods are appropriate to provide transparency and flexibility to patients, which is helpful to stimulate SDM and thereby enhance patient-centeredness. In specialist mental health care, SDM, evidence-based methods and appropriate flexibility in the treatment offered are increasingly important. Greater flexibility in the treatment offered is needed to improve SDM. The goal is to involve patients in the decision making, in dialogue with their relatives and professionals [29] using evidence-based methods. The flexibility is needed to consider options and to meet the needs of patients.

The main advantage of Method II compared to Method I was that the content of a care package can be configured to the needs of one individual patient. For this reason, the patient-centredness of Method II seems higher than of Method I. Although this result is different from what was expected, it indicates that treatment components often are not completely independent. When there are patterns in the sequence of treatment components for a specific patient group, in other words one treatment component logically follows another component, it would be helpful to create prototypes [2,4]. This explains why representatives of some centres tend to prefer Method I.

Since each care centre tends to adopt ‘generic elements’ to some extent, the findings support the prototype method. Prototypes can serve as a ‘predefined base package’ starting from which individual packages can be further specified. Modules can be process- or content-based. Chorpita et al. [21] describe these elements as coordination-oriented and content-oriented. Content modules function as manuals for treatment interventions based on guidelines. Coordination or process modules are the ‘cement holding it together’, for example, a care consultation meeting. The extent to which prototypes can be filled in with process or content modules differs per patient group.

Since this is one of the first case studies that applies modularity to specialist mental health care, it may be interesting to conduct more extensive research on several mental health care organisations, where experiences of practitioners and patients are analyzed as well to compare the customisation methods. Research should then focus on the influence of both methods on transparency, flexibility, customisation and SDM.

### 4.3. Strengths and Limitations

A theory-based case study research approach was appropriate because the current theory and practice-based knowledge are at a formative stage. Moreover, a combination of additional methods i.e., literature review, document analysis and interviews were performed. Qualitative methods were used to ask additional questions and modify in-depth discussions. Additionally, existing reports and minutes were used to provide additional insights into the organisation of treatment for data triangulation. By using different data sources, results became more reliable by triangulation. Moreover, a broad representation of respondents participated, so that certain patterns emerged from this variated sample and key themes were represented [24].

A limitation of our research is that it was a single case study conducted in one specialist mental health care organisation. However, the case study was based on multiple care centres and a variety of patient groups which created a broad view. The results can be translated with certain care to similar specialist mental health care organisations treating comparable patient groups.

## 5. Conclusions

Two methods, ‘Prototype’ and ‘Menu-based’, are suggested. Based on three characteristics, i.e., predictability of needs, availability of evidence and interconnectedness of activities in treatment, which differ between patient groups, one of these methods would be preferred. Because the prototype method with the option to add or remove modules could also offer sufficient flexibility and customisation, both methods are flexible enough to facilitate SDM aiming at customisation of treatment, which is important in current mental health care.

## Figures and Tables

**Figure 1 ijerph-18-11129-f001:**
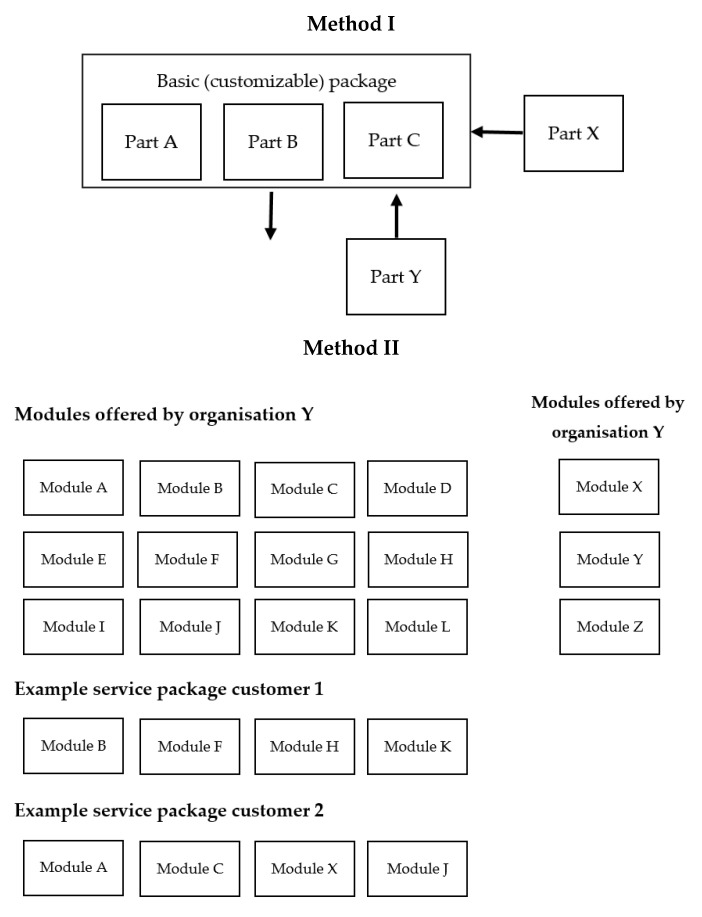
Both customisation methods: prototype (Method I) and menu-based (Method II).

**Figure 2 ijerph-18-11129-f002:**
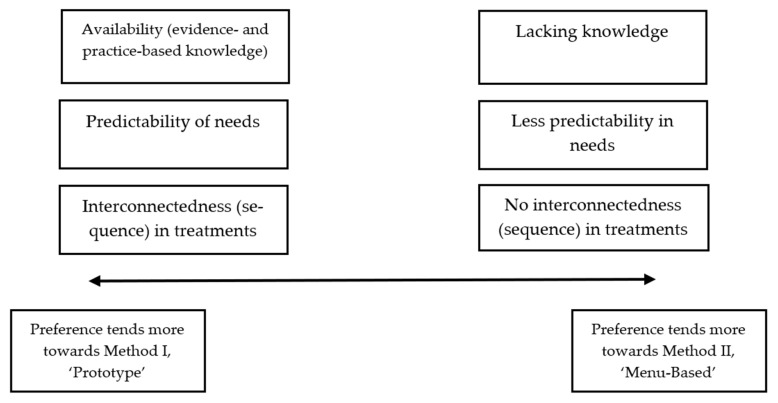
Subdivision in preference Method I ‘Prototype’ or Method II ‘Menu-based’ outlined as a conductive scale.

**Table 1 ijerph-18-11129-t001:** Review prototype and menu-based method.

	Advantages	Disadvantages
Method I: Prototype	- Integrated quality - Guidance for practitioners at the beginning of a treatment - Provides a clear framework - Supports the structuring of processes and therefore provides an overview	- Evidence-based information about the order in treatment is needed to create prototypes - Perceived as hard to assign rela-tively unpredictable patient needs in a pathway - Less freedom for the practitioner - More focused on ‘homogeneous’ patient group
Method II: Menu-based	- Because of its flexibility and the possibility to customise treat-ments for a specific patient, it offers more possibilities to cus-tomise compared to method I - Patients may experience ‘more possibilities to choose’ - It seems patient-friendly to provide a transparent overview of the offered modules from which a patient can choose	- Less overview in the treatment process of patient groups compared to method I - Illogical to split up the care process if there is an order in treatment compo-nents (based on evidence) - Unclear how to integrate coherence between treatment components - Unclear how to integrate a planned end of treatment - Patients may ask for more treat-ment because of a lot of module options - Too much information for patients

## Data Availability

The data presented in this study are available on request from the corresponding author. The data are not publicly available to ensure confidentiality.

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
