# Peer review of "Improving Customisation in Clinical Pathways by Using a Modular Perspective"

_ijerph, 2021, doi:10.3390/ijerph182111129_

Round 1
Reviewer 1 Report
I consider the document to be of great importance and of great interest at present, it presents a clear and well developed methodology; congratulations for your paper.
Author Response
We thank the reviewers for the dedicated, valuable and positive feedback on our revised manuscript.
Reviewer 2 Report
The authors revised the manuscript following the suggestions of the reviewers and the maunscript improved.
Author Response

(The authors gave the same response as above.)

Reviewer 3 Report
Thank you for giving me the opportunity to read and comment a report “Improving customization in clinical pathways by using a modular perspective”, by van Dam A., et al.
In the reviewed manuscript, the flexibility of clinical pathways in specialized health care in order to stimulate SDM have been evaluated.
This paper is well written, correctly structured with a suitable research concept, the study limitations are addressed and definitely it is of relevance to readers of the journal. However, some suggested minor changes are included in the comments given below.
- I recommend that the authors improve the quality of Figures 1 and 2.
- Strengths and limitations are usually located in the final part of the "Discussion" section.
Author Response
Please see the attachment.

This manuscript is a resubmission of an earlier submission. The following is a list of the peer review reports and author responses from that submission.
Round 1
Reviewer 1 Report
The scientific method is not clearly described, it does not mention whether it is a descriptive, correlational or multiple study; nor does it establish whether it is cross-sectional or longitudinal, the calculation of the sample should be clarified, with respect to the selection of the participants, the specific characteristics involved should also be mentioned.
The rest of the document complies with the expected requirements.
Reviewer 2 Report
This is an interesting article about possibilities to combine the conflicting approaches of standardization and patient centeredness of medical treatment. The article is based on a literature study and qualitative interviews of stakeholders. The manuscript is well written. However, some suggestions might further improve the manuscript.
1 Page 2, line 97. By using different data sources, it was intended to improve reliability of results. Whether this goal was achieved is something for the discussion. Please consider.
2 Page 4, line 130. SDM is already introduced and does not need to be introduced again.
3 Page 4, line 139. The 23 representatives are already described in the method section
Thank you for the chance to read this interesting paper
Reviewer 3 Report
The topic of the article is definitely relevant, since, as the authors themselves point out in the introduction, clinical pathsways are often not actually individualized to the specific needs of the users. By their very nature, clinical pathways tend to organize patients in a homogeneous and linear form. They, do not target patients with unique requirements and contexts and do not facilitate Shared Decision Making (SDM).
The Authors claim that they studied the extent to which flexibility in the way of working with clinical pathways in specialist mental healthcare can help to stimulate SDM aiming to enhance patient-centeredness. They evaluated modularity, a concept originating from operations management, as a tool to reconcile standardisation and customization. Modularity refers to a set of principles for managing complexity by dividing a system into standardized parts in order to create a variety of configurations to meet heterogeneous needs.
A literature search was performed using “modularity”, “modularity in (mental) healthcare provision”, “clinical pathways”, “personalization” and “customization (in healthcare”, “(mental) healthcare provision”, “patient-centeredness”, “Shared Decision Making”, combinations between those keywords 84 and focused on papers published between 2000 and 2010.
It not clear to me why the search was limited to 2010. The authors should explain why the following 10 years were not evaluated.
Subsequently, a case was evaluated through interviews, interviews etc. with various professional figures in the field of mental health and he results were coded by two researchers.
Literature review identified two approachs: ‘Prototype’ and ‘Menu-based’. In the former the customer gets a standardized treatment which can be further tailored to suit requirements. In the latter, modularity is applied via a menu of options from which a service employee combines several components in order to meet specifications of the customer.
Understandably, the literature review shows that patients appreciate being involved in treatment decisions.
Interviews with mental health operators have shown that the main limitation is given by the lack of flexibility in the use of clinical pathways. Another limitation of the therapeutic pathways is given by the fact that the alternatives to treatment are limited by the clinical pathway itself.
Authors conclude that a “menu-based” approach may be more suitable to improve therapeutic pathways.
I found it difficult to understand exactly what the interviews were conducted on. The authors report that they used a clinical case, which however is not described. Can they explain better how the interviews were developed and what questions were asked?
In addition, the 23 people interviewed will have exposed common themes and different themes. Is it possible to have a table with the coding of the main problems exposed and of the different perspectives, also with respect to the different professional qualifications?
Overall I think it is a good paper that deserves publication, but it must be explained why the bibliographic research was limited to the previous decade and the qualitative description of the data must be extended.
Another point the Authors may wish to comment is how, in their opinion, SDM should be evaluated when considering also competence to treatment in a psychiatric population. From their data it seems that competence for treatment of the patients was not a concern for the health professionals. Research has shown that informed consent in psychiatric patients is often compromised and this could be a limiting element in SDM.